# Minimax Based Fast-training Defense against Adversarial Policy in Two-player Competitive Games

## Abstract

Adversarial policies have been shown to exploit vulnerabilities in agents during two-player competitive games, significantly undermining their performance. While existing approaches model the challenge of training robust policies in such environments as the search for Nash equilibrium points in the policy space, this often leads to substantial computational overhead. In this work, we propose MM-FATROL, a novel robust policy training method grounded in the Minimax Theorem, which significantly reduces computational overhead by efficiently identifying promising policy updates. We provide a formal analysis of the speedup achieved by our method. Extensive experiments demonstrate that MM-FATROL not only enhances efficiency but also surpasses the state-of-the-art method in terms of generalization and robustness. Additionally, we discuss the limitations of our approach and the challenges that remain in developing robust policies for more complex game environments.

## 1 Introduction

Reinforcement Learning (RL) has long been a prominent area of academic research, with interest further intensified by its integration with deep learning technologies. Deep Reinforcement Learning (DRL) combines the decision-making capabilities of RL with the representational power of deep learning, enabling agents to approximate complex policy update processes through continuous interaction with the environment. As a versatile end-to-end control system, DRL has achieved, and in many cases surpassed, human expert-level performance in fields like robotic control (van Hasselt et al., 2016), autonomous driving (Liao et al., 2022), recommendation systems (Huang et al., 2021), and game AI. Notably, in the realm of game AI, DeepMind's AlphaGo (Silver et al., 2016) and AlphaGo Zero (Silver et al., 2017) have defeated top professional players in the two-player zero-sum game of Go. In 2019, DeepMind extended these successes with AlphaStar (Vinyals et al., 2019; 2017), outperforming 99.8% of human players in StarCraft II.

Many DRL applications require high levels of security and stability, such as communication flow control (Liu et al., 2021) and intelligent transportation systems (Haydari & Yilmaz, 2022). However, recent researches have revealed that DRL models are vulnerable to various attacks. It is known that deep learning models are susceptible to adversarial samples (Lin et al., 2020; Dong et al., 2018; Kurakin et al., 2017), where small perturbations to the input can lead to incorrect outputs. In the DRL setting, similar techniques can be used to influence agents into making poor decisions. Huang et al. (2017) apply adversarial examples to DRL, demonstrating that noise introduced by the FGSM method (Goodfellow et al., 2015) can cause DQN (Mnih et al., 2013) and PPO (Schulman et al., 2017) models to make erroneous decisions. Behzadan & Munir (2017) adopt the idea of transfer-based attacks , where a surrogate model predicts how input modifications will cause the victim agent to underperform.

The above adversarial attacks primarily target the deep learning component of DRL through adversarial perturbation or poisoning attacks. On the other hand, Gleave et al. (2020) introduced adversarial policy attacks, which do not directly modify the input but instead train adversarial agents to force victim agents into suboptimal actions. This approach has proven effective in environments like Mujoco (Todorov et al., 2012), offering a more realistic attack. Guo et al. (2021) further advanced

this field by reconstructing the attack objective to maximize the average expectation of the attacker's policy while minimizing the victim's average reward, achieving successful attacks in StarCraft II.

Most defense mechanisms in DRL are derived from traditional adversarial attack and defense strategies, focusing on defense against perturbation-based attacks such as adversarial training and adversary detection. For instance, Kos & Song (2017) apply adversarial training to the A3C algorithm in the Atari Pong scenario, using random samples and adversarial samples generated by FGSM. Chen et al. (2018) extended adversarial training on agents to the more complex domain of autonomous navigation. Lin et al. (2017) focused on detecting adversarial samples through predictive modeling of future observations. Despite progress in defending against perturbation-based attacks, research on defending against adversarial policy attacks remains limited. Guo et al. (2023) proposed training agents in two-player games to reach Nash equilibrium conditions, thereby ensuring a performance lower bound when under attack.

In this work, we focus on robust policy training in two-player game scenarios. Inspired by the approach of transforming robust policy training into a search for Nash equilibrium in the policy space, we propose a novel robust policy training method against adversarial policy attacks called **MiniMax** based **FA**st-training defense agains**T** adversa**R**ial p**OL**icy (MM-FATROL). Grounded in the Minimax Theorem (Cheng et al., 2014), MM-FATROL reduces computational overhead while maintaining strong robustness. Extensive experiments demonstrate that MM-FATROL significantly reduces computational costs compared to state-of-the-art methods, while also achieving superior performance across various games. Additionally, MM-FATROL exhibits stronger robustness against adversarial policy attacks. We also discuss the limitations of our algorithm and the challenges of achieving the most robust policy in arbitrary game environment.

Our main contributions are as follows:

- We propose MM-FATROL, a novel robust policy training algorithm based on the Minimax Theorem, and provide a formal analysis of its computational efficiency compared to state-of-the-art methods.
- We demonstrate through extensive experiments that MM-FATROL reduces computational overhead while maintaining top-tier performance and stronger robustness against adversarial policy attacks across various games.
- We analyze the key challenges in robust policy training and identify future directions for enhancing the defense against adversarial policy attacks.

## 2 PRELIMINARIES

### 2.1 DEEP REINFORCEMENT LEARNING

Deep reinforcement learning (DRL) integrates deep learning methods into the core principles of reinforcement learning (RL). Currently, most DRL algorithms are based on the Actor-Critic framework, where deep neural networks are employed to approximate both the policy and value functions of the agent, with the former for action selection and the latter for action evaluation. Among these algorithms, Proximal Policy Optimization (PPO) (Schulman et al., 2017) is widely recognized for its simplicity and efficiency, making it the preferred choice for most continuous control tasks. As a policy gradient algorithm, PPO updates the policy $\pi_{\theta'}$ using data from interactions with the environment under the old policy $\pi_\theta$, employing importance sampling. The objective function of PPO is given by:

$$J_{PPO}^{\theta}(\theta') = \mathbb{E}[\min(\text{clip}(r_t, 1 - \varepsilon, 1 + \varepsilon)A_\theta, r_t A_\theta)].$$

where $r_t = \frac{\pi_{\theta'}(a_t|s_t)}{\pi_\theta(a_t|s_t)}$ represents the importance sampling ratio, and $A_\theta = Q_{\pi_\theta}(s_t, a_t) - V_{\pi_\theta}(s_t)$ is the advantage function. PPO uses a clipping mechanism to limit the magnitude of policy updates, enhancing sampling efficiency while ensuring algorithm stability. To further improve training performance, the DPPO (Distributed PPO) algorithm (Heess et al., 2017) utilizes a primary network to compute gradients and update parameters, while several sub-networks collect data, significantly boosting both training efficiency and policy quality.

## 2.2 TWO-PLAYER MARKOV GAME AND NASH EQUILIBRIUM

Multi-agent reinforcement learning is often framed as a Markov game (Littman, 1994), an extension of the single-agent Markov decision process. A Markov game is typically represented as a sextuple $\mathcal{G} = (\mathcal{N}, \mathcal{S}, \{\mathcal{A}^i\}_{i=1}^n, P, \{r^i\}_{i=1}^n, \gamma)$, where $\mathcal{N} = \{1, \ldots, n\}$ is the set of players, $\mathcal{A}^i$ and $r^i : \mathcal{S} \times \prod_{i=1}^n \mathcal{A}^i \to \mathbb{R}$ represent the action space and reward function of player $i$, respectively. In two-player Markov games, $n = 2$, and the joint action at time $t$ is $\boldsymbol{a}_t = (a_t^i, a_t^{-i})$, where $a_t^{-i}$ represents the opponent's action. Both players receive an immediate reward $r_t^i = r^i(s_t, \boldsymbol{a}_t)$ (if $r_t^i + r_t^{-i} = 0$ holds for any $t$, then it is a zero-sum game), and the environment transitions to the next state $s_{t+1} \sim P(\cdot|s_t, \boldsymbol{a}_t)$. The state value function and the action value function for player $i$ are given by:

$$V_{\boldsymbol{\pi}}^i(s) = \mathbb{E}_{a_t^i \sim \pi^i, a_t^{-i} \sim \pi^{-i}} \left[ \sum_{t=k}^{\infty} \gamma^{t-k} r^i(s_t, \boldsymbol{a}_t) \Big| s_k = s \right],$$

$$Q_{\boldsymbol{\pi}}^i(s, \boldsymbol{a}) = r^i(s, \boldsymbol{a}) + \gamma \cdot \mathbb{E}_{s' \sim P(\cdot|s, \boldsymbol{a})} \left[ V_{\boldsymbol{\pi}}^i(s') \right].$$

In this setup, a rational player $i$ aims to maximize his cumulative expected reward $U^i$. Given the opponent's policy $\pi^{-i}$, player $i$ will always select the Best Response (BR) policy $\pi^i$ that maximizes his reward. This is formalized as $\mathrm{BR}^i(\pi^{-i}) = \{\pi^i \in \Delta(\mathcal{A}^i) | U_{\pi^i, \pi^{-i}}^i = \max_{\mu \in \Delta(\mathcal{A}^i)} U_{\mu, \pi^{-i}}^i\}$. When both players adopt policies that are best responses to one another, the policy combination forms a Nash equilibrium (NE) (Nash, 1950), formally defined as $\forall i \in \mathcal{N}, \pi_*^i \in BR(\pi_*^{-i})$, where $\boldsymbol{\pi}_* = (\pi_*^i, \pi_*^{-i})$ is a Nash equilibrium. The key property of a Nash equilibrium is that neither player can improve their payoff by unilaterally changing their policy.

For finite two-player zero-sum games, von Neumann's Minimax Theorem guarantees the existence of a Nash equilibrium. Specifically, for any player $i$, the following holds:

$$\max_{\pi^i} \min_{\pi^{-i}} U_{\pi^i, \pi^{-i}}^i = \min_{\pi^{-i}} \max_{\pi^i} U_{\pi^i, \pi^{-i}}^i.$$

Shapley (1953) extended this result to Markov games, proving that for finite state and action space, there exists a pair of stationary policies that satisfy the Nash equilibrium property.

## 2.3 THE PATROL METHOD

Guo et al. (2023) discovered that in two-player zero-sum games, training robust policies for both players can be framed as the search for a Nash equilibrium in the policy space. From a game-theoretic perspective, the policy pair at the Nash equilibrium represents a set of robust policies capable of maintaining a performance lower bound under any adversarial policy attack. Since neither player can improve their payoff by unilaterally altering their policy at the Nash equilibrium, even if one player's policy is replaced with an adversarial policy, the attacker cannot achieve a better outcome, thereby ensuring the victim's performance does not degrade.

Based on that insight, the PATROL method was designed to train robust policies. It initializes a policy pool for both players, consisting of $K$ pairs of policies $(\pi_k^1, \pi_k^2)_{k=1:K}$. In each iteration, all policies are updated. For player $i$ in the $j$-th iteration, the strongest opponent policy $\pi_{j,v}^{-i}$ that minimizes the payoff of $\pi_{j,k}^i$ is selected from the opponent's policy pool $\{\pi_{j,\tilde{k}}^{-i}\}_{\tilde{k}=1:K}$. DPPO is then used to update $\pi_{j,k}^i$ against this opponent. After multiple iterations, the optimal policy is chosen based on the highest average winning rate from the final payoff matrix.

# 3 THE PROPOSED METHOD

## 3.1 PROBLEM SETUP

**Adversarial Policy Attack.** In this context, adversarial attacks target agents that have already been trained within a two-player competitive environment. The attacker selects one of the player agents as the victim, fixing the victim's policy, $\pi^i$, which effectively transforms the original Markov game process into a Markov decision process (MDP). The attacker's goal is to find an optimal attack policy, $\pi_\alpha$, within this MDP that minimizes the victim's cumulative reward. During training, the

attacker can observe the victim's actions but does not have access to any white-box information about the victim's model, such as its structure or parameters. Additionally, the attacker cannot manipulate the game environment or interfere with the feedback provided by the environment to the agent. This scenario models real-world adversarial attacks on policies trained using deep reinforcement learning, such as attacks on autonomous vehicles that degrade their performance in navigation or obstacle avoidance, posing serious safety risks.

**Assumptions for Defenders.** Similar to the attacker, the defender is also unable to manipulate the game environment. Furthermore, the defender cannot interfere with the attacker's training process or predict the attacker's policy in advance. This restriction means that, without disrupting the attacker's intentions, the defender cannot engage in numerous confrontations with adversarial policies to gather training data. Therefore, the defender cannot perform specific defenses through adversarial retraining (Guo et al., 2023).

**Objective of Our Method.** The goal of this work is to develop robust policy pairs for both players in the game, ensuring that each party's policy maintains a certain lower bound of performance even in the presence of adversarial policy attacks. Additionally, the trained policies should possess generalization capabilities, performing well not only under attack but also in standard, non-adversarial scenarios.

## 3.2 THE PROPOSED ALGORITHM

In this work, we integrate the Minimax Theorem into the framework of PATROL to further enhance the performance. The procedure is outlined in Algorithm 1, with three key improvements detailed below:

**Select Promising Policies for Updates.** The core idea of the PATROL algorithm is to search for a Nash equilibrium in the policy space, ensuring the victim agent's performance be above a lower bound under adversarial policy attacks. Traditionally, this involves updating all policies against their strongest opponents in each iteration, gradually converging toward the Nash equilibrium. However, this approach can lead to significant increase on computational overhead. Instead of that, we identify the most promising policy combinations for training in each iteration, without expending substantial resources on unnecessary computations.

Using the Minimax Theorem, we can identify these target policies. For instance, player $i$'s maximin value is given by $\mu = \max_{\pi^i} \min_{\pi^{-i}} U^i_{\pi^i, \pi^{-i}}$, with the corresponding policy combination $(\pi^i_\mu, \pi^{-i}_\mu) = \arg\max_{\pi^i} \arg\min_{\pi^{-i}} U^i_{\pi^i, \pi^{-i}}$. This ensures that $\pi^i_\mu$ is the most robust policy in player $i$'s pool, guaranteeing a payoff of at least $\mu$ when facing an unknown opponent. Simultaneously, $\pi^{-i}_\mu$ is the strongest adversarial policy for player $i$, as it leads to the lowest payoff $\mu$ for player $i$. Similarly, the minimax value $\nu = \min_{\pi^{-i}} \max_{\pi^i} U^i_{\pi^i, \pi^{-i}}$ corresponds to the policy combination $(\pi^i_\nu, \pi^{-i}_\nu) = \arg\min_{\pi^{-i}} \arg\max_{\pi^i} U^i_{\pi^i, \pi^{-i}}$, where $\pi^{-i}_\nu$ is the most adversarial policy for player $i$, and $\pi^i_\nu$ is its strongest counter.

Thus, in each iteration of MM-FATROL, we select $\pi^i_\mu$ from player $i$'s pool as the most worthwhile policy for training, and $\pi^{-i}\mu$ as the opponent's policy to assist in updating $\pi^i_\mu$. For player $-i$, we use $\pi^i_\nu$ as the fixed opponent policy while updating $\pi^{-i}_\nu$ using DPPO. This approach offers a more efficient and targeted strategy for policy updates.

**Design Update Windows.** In each iteration, only one policy from each policy pool is selected for updating , which may introduce some bias in the search direction and limit exploration. To address this, we propose the concept of update windows to correct the update direction. We distinguish between two types of updates: "minimax updates" (updating only the selected promising policy) and "full updates" (updating all policies in the pool). Specifically, we define $c$ iterations as one update window. During each window, we perform one full update followed by $j$ minimax updates in a cyclic manner. As we enter the next update window, the number of minimax updates, $j$, is incremented by a parameter $a$ (referred to as "acceleration"), with the condition that $j \in [0, m]$, where $m$ is the "speed limit". By interspersing full updates between minimax updates, we strike a balance between expanding the search range in the policy space and refining the update direction, reducing the risk of suboptimal outcomes.

---

**Algorithm 1** MM-FATROL

---

**Input:** Number of iterations $I$, size of policy pool $K$, size of window $c$, acceleration $a$, limiting-velocity $m$.
1: Initialize $K$ pairs of policies $(\pi_k^1, \pi_k^2)_{k=1:K}$.
2: Initialize iteration $i = 1$ and speed $s = 0$.
3: **for** $w \leftarrow 1, ..., I/c$ **do**
4:     **for** $p \leftarrow 1, ..., c/(1+s)$ **do**
5:         Do a full update.
6:         **for** $q \leftarrow 1, ..., s$ **do**
7:             Find $(\pi_{\mu i}^1, \pi_{\mu i}^2) = \text{argmax}_{\pi^1} \text{argmin}_{\pi^2} U_{\pi^1, \pi^2}^1$
                and $(\pi_{\nu i}^1, \pi_{\nu i}^2) = \text{argmax}_{\pi^2} \text{argmin}_{\pi^1} U_{\pi^1, \pi^2}^2$.
8:             Update $\pi_{\mu i}^1$ against $\pi_{\mu i}^2$ using DPPO.
9:             Update $\pi_{\nu i}^2$ against $\pi_{\nu i}^1$ using DPPO.
10:            $i \leftarrow i + 1$
11:        **end for**
12:        $s \leftarrow \min(s + a, m)$.
13:    **end for**
14: **end for**
**Output:** $(\pi_{\mu I}^1, \pi_{\nu I}^2)$.

---

**Select the Optimal Policies.** In PATROL, once all iterative updates of the policy pool are completed, the policies with the highest mean payoff from both parties' policy pools are selected as the optimal output. However, in the final stage of MM-FATROL, following the Minimax Theorem, we select policies corresponding to the maximin values for both players as the optimal outcomes. A detailed analysis of this approach is provided in Section 5.1.

### 3.3 THEORETICAL GUARANTEE

**Convergence to NE.** The convergence of PATROL to a Nash Equilibrium was established by Guo et al. (2023), demonstrating that all policy combinations ultimately converge to an NE. Our method, MM-FATROL, builds on this foundation by iteratively updating the policy pairs, thereby theoretically ensuring convergence to the NE as well.

**Reduction Ratio of Computational Overhead.** We derive a lower bound on the computational overhead reduction ratio of MM-FATROL compared to PATROL, as stated in Theorem 1. The proof for this theorem is included in Appendix A.

**Theorem 1.** *Let $o$ denote the computational overhead required for a single parameter update of any policy $\pi$. MM-FATROL guarantees a lower bound on the reduction ratio of computational overhead over PATROL as follows:*

$$\eta > 1 - \frac{c(m+a)((K+2)m + 2K) + 2(ar - mc)(m+K)}{2Kan(m+1)},$$

*where $c, a, m, k$ are algorithm parameters, while $r$ and $n$ represent the number of iterations required for MM-FATROL and PATROL to converge to the NE, respectively.*

## 4 EXPERIMENTS

### 4.1 EXPERIMENTAL SETUP

**Environment setup.** We select two types of game environments to showcase the advantages of MM-FATROL over baseline methods.

- **Euclidean Games.** In this setting, both players control the x- and y-coordinates, aiming to achieve opposing values of the function $f(x, y)$. Player 1 controls $x$, with the objective of minimizing $f(x, y)$, resulting in a value function of $-f(x, y)$. Conversely, player 2 controls $y$, aiming to maximize $f(x, y)$, with a value function of $f(x, y)$. For our experiments, we adopt benchmarks

from Guo et al. (2023) and evaluate six different Euclidean games with varying properties: two with convex-concave value functions (ID: 1, 2), two with asymmetric action spaces (ID: 3, 4), and two with non-convex non-concave value functions (ID: 5, 6). Each type includes a simple game with a smaller domain and a more complex game with a larger domain.

- **MuJoCo Games.** We also select four games from the MuJoCo platform: YouShallNotPass and KickAndDefend (with asymmetric action spaces), as well as SumoHumans and SumoAnts (with symmetric action spaces). These environments feature continuous state/action spaces and complex DRL training environments with non-concave and non-convex value functions. Among these games, only YouShallNotPass is a zero-sum game, while the other three are general-sum games.

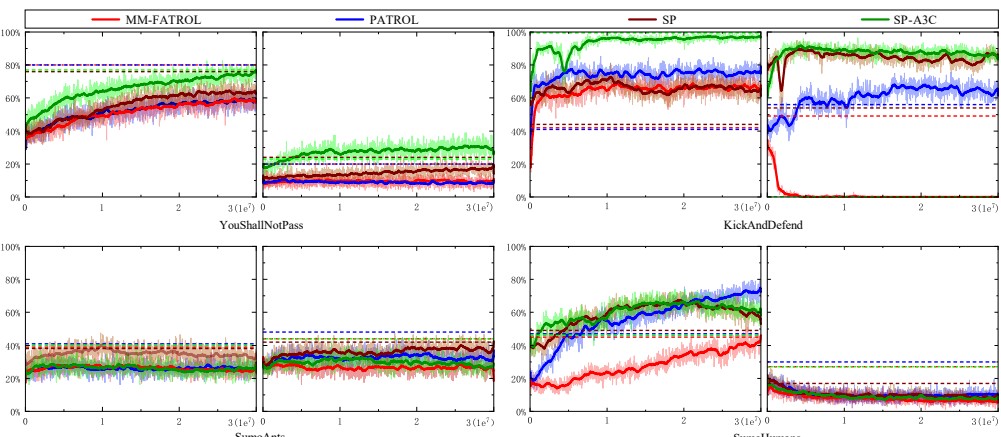

Figure 1: Robustness comparison between MM-FATROL and three baselines against adversarial policy attacks. The x-axis represents the training time steps of the adversarial policy, while the y-axis indicates its winning rate. The darker solid lines show the average winning rates of the adversarial policies during training, and the lighter bands reflect the variations between the maximal and minimal winning rates. The dotted line represents the winning rate of the victim's original opponent trained against the current victim policy.

**Hyper-parameters of MM-FATROL.** MM-FATROL has four key hyper-parameters: policy pool size $K$, window size $c$, acceleration $a$, and speed limit $m$. In all experiments, $K$ is set to 4. For acceleration, $a$ is set to 1 in the Euclidean games and 3 in the MuJoCo environments. The speed limit $m$ is unrestricted in Euclidean games with concave-convex value functions, but it is set to 10 in other environments. For window size $c$, it is set to 10 for Euclidean games with concave-convex value functions, while for non-concave non-convex Euclidean games, $c$ is set to 100 for the simple one and 150 for the other. In MuJoCo environments, $c$ set to 300.

**Baselines.** To evaluate the performance of MM-FATROL, we compare it against PATROL, the current state-of-the-art method for this problem. Additionally, we introduce two other baseline methods in the MuJoCo experiments to assess the generalization and robustness of our approach: self-play (SP) and self-play-A3C (SP-A3C). The SP and PATROL baselines use the same settings as in the original papers, except for the policy pool size $K$, which is set to 4 for PATROL. The SP-A3C algorithm is a variant of SP, using A3C instead of PPO for policy updates, with other hyper-parameters remaining unchanged. We will discuss the rationale behind the selection of $K$ in Section 4.2.

## 4.2 EXPERIMENT RESULTS

**Reduction of Computational Overhead.** Tables 1 and 2 compare the runtime of MM-FATROL and PATROL in both Euclidean games and MuJoCo environments. It is evident that MM-FATROL consistently requires significantly less training time than PATROL across all game settings. Notably, in Euclidean game (3), the reduction in computational overhead achieved by MM-FATROL reaches up to 55.6%. Across all Euclidean games, reductions are substantial, with the smallest reduction observed at 37.3%. For games featuring concave-convex value functions, the reduction is even more pronounced, typically exceeding 10% compared to those without such functions. In the MuJoCo environments, however, the complexity of the policy space is much greater than that of Euclidean

games, resulting in a declined overall improvement. Nevertheless, the reductions generally exceed 10%, with the highest being 25.5% in the SumoAnts scenario. These results demonstrates that our method effectively reduces the computational overhead associated with searching for NE points in the policy space, particularly in simpler game environments.

Table 1: The runtime comparison between MM-FATROL and PATROL in Euclidean games. Each setup was executed 5 times, with the average runtime reported.

| ID | Value function | Domains | NE | Runtime (h) | | Reduction |
|----|----------------|---------|-----|-------------|------------|-----------|
| | | | | PATROL | MM-FATROL | |
| 1 | $x^2 - y^2 - 2x$ | [-2,2] | (1,0) | 1.9 | 1 | 47.4% |
| 2 | $x^2 + 2xy - 4y^2 + 10x$ | [-50,50] | (-4,-1) | 15 | 7 | 53.3% |
| 3 | $x^2 - 2y^2 - 2xy - 6x$ | [-5,5],[-4,4] | (2,-1) | 4.5 | 2 | 55.6% |
| 4 | $x^2 + 4xy - 2y^2 + 24x$ | [0,50], [-50,0] | (-4,-4) | 9.2 | 4.5 | 51.1% |
| 5 | $x^2 y^2 - xy$ | [-2,2] | (0,0) | 13.3 | 8 | 39.8% |
| 6 | $x^3 - 9x^2 - 2y^2 x^3$ | [-50,50] | (6,0) | 8.3 | 5.2 | 37.3% |

**Robustness of MM-FATROL.** Figure 1 illustrates the winning rates of the adversarial policy in four MuJoCo game environments when hacking the policies trained by MM-FATROL and three other methods. In every game, our proposed method achieves the lowest winning rate for adversarial policy attacks. This is especially notable in the three general-sum games, where our method significantly outperforms the baselines in terms of robustness. Particularly in the KickAndDefend game, after training with MM-FATROL, player 2's agent can nearly completely defend against adversarial policy attacks, demonstrating robustness that is far superior to the other three methods. Further investigation reveals that because KickAndDefend is a general-sum game, there are scenarios where the game can end in a draw. The policy trained with MM-FATROL can, at worst, force a draw against adversarial attacks, ensuring that adversarial policies cannot defeat our trained agent. However, it is important to note that player 1's agent in KickAndDefend does not guarantee a winning rate against adversarial policy attacks that is lower than that of its origional opponent, as indicated by the higher winning rate of adversarial attack shown in Figure 1. Nevertheless, even without guaranteeing an ideal lower bound of performance against attacks, our method still exhibits stronger robustness compared to the baselines.

**Generalization of MM-FATROL.** Table 3 presents the winning rates of policies obtained from the four training methods. A comparison of the data within each row shows that in the KickAndDefend game, MM-FATROL clearly outperforms the other three baseline methods. In the other three games, both MM-FATROL and PATROL have their share of victories, and both significantly outperform the performance of SP and SP-A3C algorithms. These results indicate that our method exhibits high generalizability, maintaining the highest level of winning rates even when faced with agents trained through non-adversarial methods.

**Analysis on Pool Size $K$.** In PATROL, the researchers set $K = 2$ based on their tests of PATROL against self-play in YouShallNot-Pass game with $K$ values in $\{1, 2, 3\}$, where they found that the winning rates for $K = 2$ and $K = 3$ were similar but both were noticeably higher than for $K = 1$. However, when we tested the PATROL's sensitivity to $K$, we observed instability, even in the simple non-convex and non-concave Euclidean game when $K = 2$. In the complex non-convex and non-concave Euclidean game, convergence to NE points was often unattainable. This issue was significantly alleviated when we increased the pool size to $K = 4$, resulting in more stable convergence and reduced fluctuations in the results. We believe that training robust policies essentially involves searching for NE points within the policy space, and both PATROL and MM-FATROL provide accurate guidance for this search. However, the effectiveness of the search is

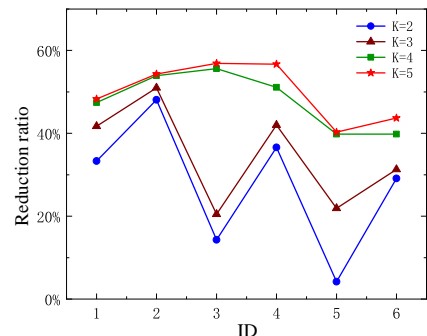

Figure 2: Reduction ratio of computational overhead achieved by MM-FATROL compared to PATROL across various $K$ values in Euclidean games.

influenced by the initial policies and especially the pool size. When the pool is too small, the search tends to fall into suboptimal or unstable states, leading to fluctuating outcomes.

We also conducted sensitivity experiments on the value of $K$ for MM-FATROL in Euclidean games, and the results are shown in Figure 2. It's evident from the figure that within the same game, the larger the value of $K$, the higher the proportion of computational overhead that our method can reduce.

Table 2: Comparison of the runtimes between MM-FATROL and PATROL in MuJoCo games. Each setup was executed 5 times and the average runtime is reported.

| MuJoco game | Runtime (h) | | Reduction |
|---|---|---|---|
| | PATROL | MM-FATROL | |
| YouShallNotPass | 470 | 385 | 18.1% |
| KickAndDefend | 620 | 491 | 20.8% |
| SumoHumans | 242 | 212 | 12.4% |
| SumoAnts | 400 | 298 | 25.5% |

Table 3: The winning rates of policies trained by MM-FATROL and other baseline methods in MuJoCo games. Method_$i$ represents the policy of player $i$ trained using the respective method. Each battle was run 1200 times and the average runtime is reported. MM refers to MM-FATROL and A3C denotes SP-A3C.

| Mujoco game | MM_1 vs. | | | | MM_2 vs. | | | |
|---|---|---|---|---|---|---|---|---|
| | MM_2 | PATROL_2 | SP_2 | A3C_2 | MM_1 | PATROL_1 | SP_1 | A3C_1 |
| YouShallNotPass | 20% | 19% | 28% | 35% | 80% | 81% | 83% | 90% |
| KickAndDefend | 49% | 57% | 94% | 88% | 42% | 49% | 50% | 95% |
| SumoHumans | 27% | 29% | 36% | 35% | 45% | 45% | 69% | 61% |
| SumoAnts | 44% | 45% | 52% | 46% | 39% | 38% | 49% | 43% |
| Mujoco game | PATROL_1 vs. | | | | PATROL_2 vs. | | | |
| | MM_2 | PATROL_2 | SP_2 | A3C_2 | MM_1 | PATROL_1 | SP_1 | A3C_1 |
| YouShallNotPass | 19% | 20% | 26% | 34% | 81% | 80% | 82% | 87% |
| KickAndDefend | 47% | 56% | 90% | 91% | 42% | 41% | 48% | 93% |
| SumoHumans | 28% | 30% | 32% | 30% | 49% | 47% | 73% | 66% |
| SumoAnts | 46% | 48% | 52% | 47% | 40% | 41% | 46% | 41% |
| Mujoco game | SP_1 vs. | | | | SP_2 vs. | | | |
| | MM_2 | PATROL_2 | SP_2 | A3C_2 | MM_1 | PATROL_1 | SP_1 | A3C_1 |
| YouShallNotPass | 17% | 18% | 24% | 35% | 72% | 74% | 76% | 81% |
| KickAndDefend | 47% | 50% | 54% | 71% | 6% | 7% | 44% | 88% |
| SumoHumans | 14% | 17% | 17% | 18% | 26% | 30% | 49% | 43% |
| SumoAnts | 36% | 36% | 42% | 38% | 32% | 32% | 38% | 36% |
| Mujoco game | A3C_1 vs. | | | | A3C_2 vs. | | | |
| | MM_2 | PATROL_2 | SP_2 | A3C_2 | MM_1 | PATROL_1 | SP_1 | A3C_1 |
| YouShallNotPass | 10% | 13% | 19% | 23% | 65% | 66% | 65% | 77% |
| KickAndDefend | 0% | 1% | 1% | 0% | 12% | 9% | 29% | 100% |
| SumoHumans | 19% | 22% | 24% | 27% | 33% | 34% | 57% | 46% |
| SumoAnts | 42% | 43% | 46% | 44% | 40% | 39% | 43% | 40% |

## 5 DISCUSSION

### 5.1 THE ESSENCE OF ROBUST POLICIES

The convergence proof of PATROL for NE is predicated on the assumption that an NE exists within the policy space, a condition met in finite two-player zero-sum games. However, in many game environments, particularly those with continuous state and action spaces, the existence of NE is not guaranteed. According to the Minimax Theorem, an NE is a special case where the minimax and maximin values coincide. Thus, for the majority of two-player games operating in continuous spaces, the robust policy training process modeled by PATROL requires extension.

Addressing the core question, in a two-player competitive game, the most robust policy is to maximize their minimum achievable payoff against any opponent's policy. This aligns with the policy that yields the maximin values within their payoff space. For instance, consider the value function $f(x, y) = (x^2 - 1)^2 - (y - x)^2$ in an Euclidean game, where the action space is defined as

$x, y \in [-2, 2]$, as shown in Figure 3a, this game does not possess a global NE. However, as illustrated in Figure 3c, the value function for player 2 identifies point $C$ as its maximin point. Thus, if player 2 adopts the policy $y = 0$, they can guarantee a payoff corresponding to point $C$, regardless of player 1's chosen policy $x = \tilde{x}$. For any alternative policy of player 2 $\tilde{y} \neq 0$, player 1 can always ensure that player 2 receives a lower payoff than that from policy $y = 0$. In other words, $y = 0$ is the most robust policy for player 2. Similarly, as shown in Figure 3b, since player 1's value function is $-f(x, y)$, the most robust policies for player 1 corresponds to the minimax points ($A$ and $B$), specifically $x = \pm 1$.

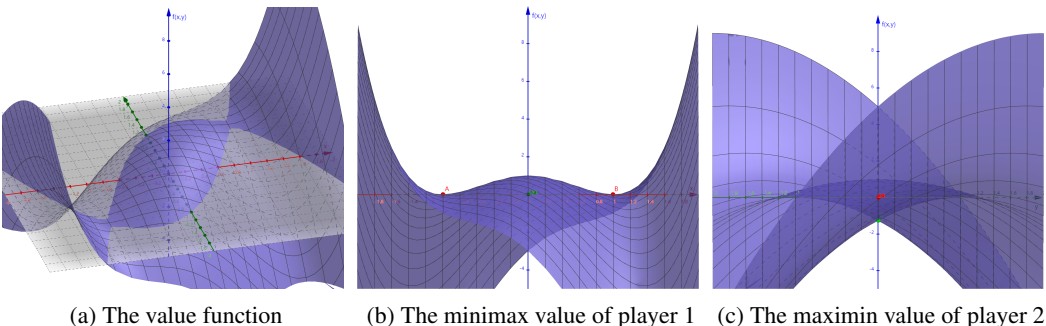

(a) The value function      (b) The minimax value of player 1    (c) The maximin value of player 2

Figure 3: Illustration of a policy space lacking a global Nash equilibrium.

## 5.2 LIMITATIONS AND CHALLENGES

As discussed in Section 5.1, in complex game environments with continuous state and action spaces where the existence of NE cannot be guaranteed, the true update direction towards a robust policy corresponds to the maximin value for each player. However, in the absence of NE points, the strongest opponent of any player's robust policy may not be the other player's robust policy. This can result in a scenario where one player's policy converges to the optimal solution while the other player does not, leading to continuous updates in the latter's policy pool without the chance to confront their theoretical strongest opponent, and thus unable to converge to the robust policy forever.

Taking the Euclidean game illustrated in Figure 3 as an example, player 1's minimax policy combinations are $(1, 1)$ and $(-1, -1)$, while player 2's optimal policy is $(\sqrt{\frac{3}{2}}, 0)$. Given that player 2's value function is a fourth-degree polynomial in $x$ and a second-degree polynomial in $y$, player 2's policy pool converges to its robust policy, specifically $y = 0$, more rapidly than player 1's. Consequently, player 1 loses the opportunity to train against the strongest opponent $y = \pm 1$ corresponding to their robust policy. As a result, player 1's policies can only train against a fixed opponent of $y = 0$ during subsequent updates, ultimately leading to convergence at $x = \sqrt{\frac{3}{2}}$, which is not actually player 1's most robust policy. Therefore, a critical challenge for enhancing robustness in future work will be ensuring that policies have the opportunity to train against their theoretical strongest opponents within the global policy space.

## 6 CONCLUSION

In this work, guided by the Minimax Theorem, we proposed MM-FATROL, a robust policy training method built on the PATROL framework. Extensive experiments demonstrated that MM-FATROL not only significantly reduces computational overhead but also maintains strong policy generalization and exhibits greater robustness compared to the state-of-the-art method. Additionally, we analyzed the limitations of existing robust policy training methods in the face of adversarial policy attacks, and outlined key challenges that must be addressed to further enhance robustness. For future work, we aim to tackle these challenges by exploring adaptive adjustments to the size of each party's policy pool based on game environment characteristics or by maintaining separate pools for the strongest opponents of each player. These directions will drive further advancements in robust policy training.

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

## A  PROOF OF THEOREM 1

**Theorem 1.** *Let o denote the computational overhead required for a single parameter update of any policy $\pi$. MM-FATROL guarantees a lower bound on the reduction ratio of computational overhead over PATROL as follows:*

$$\eta > 1 - \frac{c(m+a)((K+2)m+2K) + 2(ar-mc)(m+K)}{2Kan(m+1)},$$

*where $c, a, m, k$ are algorithm parameters, while $r$ and $n$ represent the number of iterations required for MM-FATROL and PATROL to converge to the NE, respectively.*

*Proof.* For PATROL, executing $n$ iterations implys that all policies of both players are updated $n$ times, resulting in a total cost of $O_F = 2Kon$. In the case of MM-FATROL, during the $r$ iterations of the algorithm, both "full update" and "minimax update" are present simultaneously, and the cost for one iteration of the former is $2Ko$ while the latter incurs a cost of $2o$. The algorithm begins with an "acceleration phase" comprising $(\frac{m}{a}+1)c$ iterations, of which $\sum_{i=0}^{m/a} \frac{c}{1+ia}$ iterations perform a "full update". Since the function $f(x) = \frac{1}{x}$ is a convex function for $x > 0$, for any $0 < x_1 < x_2 < x_3 < x_4$ satisfying $x_1 + x_4 = x_2 + x_3$, we have $f(x_1) + f(x_4) > f(x_2) + f(x_3)$. Using Gaussian Summation to handle the summation term above yields

$$\sum_{i=0}^{m/a} \frac{c}{1+ia} < \frac{1}{2}(\frac{m}{a}+1)(1+\frac{1}{m+1})c. \tag{1}$$

Then, by substituting inequality 1, we get the total computational overhead for the "acceleration phase" as

$$O_1 = 2Ko\sum_{i=0}^{m/a} \frac{c}{1+ia} + 2o((\frac{m}{a}+1)c - \sum_{i=0}^{m/a} \frac{c}{1+ia})$$
$$< \frac{oc(m+a)(K(m+2)+m)}{a(m+1)}.$$

The latter part of the algorithm constitutes a "stable phase" involving $r - \frac{m}{ac}$ iterations, where the proportion of "full update" is $\frac{1}{m+1}$, and the remainder consists of "minimax update". Hence, we gain the total computational overhead for the "stable phase" as

$$O_2 = 2Ko\frac{r - \frac{m}{a}c}{m+1} + 2o\frac{m(r - \frac{m}{a}c)}{m+1} = \frac{2o(ar - mc)(m+K)}{a(m+1)}.$$

Combining the two phases, the total computational overhead of MM-FATROL satisfies

$$O_M = O_1 + O_2 < \frac{co(m+a)((K+2)m+2K) + 2o(ar - mc)(m+K)}{a(m+1)}.$$

Finally, we can conclude that the reduction ratio has the following lower bound

$$\eta = \frac{O_F - O_M}{O_F} > 1 - \frac{c(m+a)((K+2)m+2K) + 2(ar - mc)(m+K)}{2Kan(m+1)}.$$

$\square$

