# OpenReview forum: "Minimax Based  Fast-training Defense against Adversarial Policy in Two-player Competitive Games"
_ICLR.cc/2025/Conference — Submitted to ICLR 2025_

### Official Review · Reviewer_CYCq · 2024-10-25

**Soundness:** 2
**Presentation:** 2
**Contribution:** 2
**Rating:** 5
**Confidence:** 3

**Summary:**

The paper introduces MM-FATROL, a robust policy training method build on PATROL framework for two-player competitive games. It uses a minimax-based selective update mechanism to only update the most impactful policies, reducing computational costs. Theoretical guarantees are provided for reduced computational complexity, and experiments show superior performance in terms of both efficiency and robustness against adversarial attacks.

**Strengths:**

1.	The MM-FATROL algorithm introduces a minimax-based selective update mechanism, which significantly reduces computational costs compared to the original PATROL framework.
2.	This paper provides theoretical guarantees that MM-FATROL not only maintains robustness but also achieves lower computational complexity than PATROL.
3.	Experimental results show that MM-FATROL outperforms baseline methods.

**Weaknesses:**

Please refer to Questions.

**Questions:**

I have a question regarding the objective of this paper, which appears to be the development of robust policy pairs for both players in the game. From the description throughout the paper, this objective seems akin to computing an equilibrium strategy profile for the game. Does this mean that the equilibrium strategy is considered the most robust strategy?

If so, could the authors clarify why they did not adopt existing equilibrium-finding algorithms, such as the PSRO algorithm, which also maintains a policy pool for each player and iteratively computes best-response strategies?

If not, could the authors explain what is meant by “robustness” in this context and how it is measured? I may have missed some important details in the paper, so please feel free to correct any misunderstandings.

Q1: The paper discusses developing robust policies, which seems closely related to finding equilibrium strategies. Could the authors clarify the distinction between robustness and equilibrium in this context?

Q2: This paper introduces a minimax-based selective update mechanism, but the explanation seems unclear. Could the authors elaborate on how the minimax strategy is computed, specifically in Line 7 of Algorithm 1? Additionally, in Algorithm 1, the variable $i$ appears to keep increasing without resetting. Is this intentional, or is there a step where it should be reset?

---

### Official Review · Reviewer_HgHq · 2024-10-29

**Soundness:** 1
**Presentation:** 2
**Contribution:** 2
**Rating:** 3
**Confidence:** 4

**Summary:**

By following previous approaches, this paper models the challenge of training robust policies in such environments as the search for Nash equilibrium points in the policy space, this often leads to substantial computational overhead. They then propose MM-FATROL, a novel robust policy training method grounded in the Minimax Theorem used in the existing algorithm FATROL, which reduces computational overhead by efficiently identifying promising policy updates. Their experiments demonstrate that MM-FATROL shows promising results in terms of generalization and robustness.

**Strengths:**

They propose MM-FATROL, a robust policy training method grounded in the Minimax Theorem, which reduces computational overhead by efficiently identifying promising policy updates. Their experiments demonstrate that MM-FATROL shows promising results in terms of generalization and robustness.

**Weaknesses:**

-The novelty is limited because the proposed algorithm slightly modifies the existing algorithm FATRO by focusing on the best strategy instead of all strategies.

-The challenge is unclear. What is the bottleneck of FATRO? How does the proposed algorithm overcome this challenge? It seems that solving a maxmin problem is much harder than solving a min problem. Then, it is unclear how the proposed algorithm reduces the computation overhead.

-This paper overstated that the proposed algorithm guarantees converging to a Nash equilibrium. However, that is impossible in general games.
As the authors mentioned in Line 374, “We believe that training robust policies essentially involves searching for NE points within the policy space, and both PATROL and MM- FATROL provide accurate guidance for this search. However, the effectiveness of the search is influenced by the initial policies and especially the pool size. When the pool is too small, the search tends to fall into suboptimal or unstable states, leading to fluctuating outcomes.” This discussion has shown that the converging to a Nash equilibrium is not guaranteed.

-The proposed algorithm is still similar to the self-play algorithm (i.e., only consider best response to the opponent’s strategy). As we know self-play algorithm cannot guarantee converging to a Nash equilibrium.


-The presentation should be better. Some notations are used without definition, e.g., \pi, U_\pi, \eta, and so on.

**Questions:**

see the above

---

### Official Review · Reviewer_wa53 · 2024-11-01

**Soundness:** 2
**Presentation:** 2
**Contribution:** 2
**Rating:** 3
**Confidence:** 3

**Summary:**

The subject of the paper is adversarial policies in two-player games (both zero-sum and general sum). Adversarial policies are those which try to minimize the payoff of the defender. The authors introduce new ideas for computing these based on a prior iterative approach that converges to an NE in zero-sum games, namely they update the policies that perform strongest in the current minimax optimization and they have hyperparameters to govern how often these "strong policy updates" happen relative to updates of all policies. During training, more and more strong policy updates are done for each full update.

The authors carry out empirical analysis of their algorithm in four games and find that it achieves more robustness in less computational time.

**Strengths:**

- The algorithm fits in with the existing framework
- Empirical results are positive

**Weaknesses:**

- The method adds additional hyperparameters which are tuned per game. The value of these hyperparameters appears to matter. There is relatively limited discussion of hyperparameter tuning. The computational time benefits seem like they could be outweighed by the cost of performing additional hyperparameter tuning.
- The theoretical proof of the lower bound of reduced computational complexity does not seem to discuss how the number of iterations required MM-FATROL could be changed relatively to PATROL. There is also no comparison of the difference predicted by theory compared to what is seen in practice.

Taken together, I don't believe the paper should be accepted. The algorithmic contributions are not significant enough when combined with the amount of empirical validation. The theory does not seem to add much to the paper in its current form. Discussion is relatively surface-level (I do not get what we are supposed to learn from 5.1 and 5.2).

**Questions:**

1. How were hyperparameters other than K set? How sensitive is the algorithm to these hyperparameters?

---

### Official Review · Reviewer_2JXN · 2024-11-03

**Soundness:** 2
**Presentation:** 3
**Contribution:** 2
**Rating:** 3
**Confidence:** 4

**Summary:**

This paper addresses the challenge of training robust policies in two-player adversarial games, where current approaches, often based on Nash equilibrium, involve high computational costs. The authors propose MM-FATROL, a robust policy training method grounded in the Minimax Theorem, which reduces computational overhead by efficiently identifying promising policy updates.

**Strengths:**

1.	The paper uses Nash equilibrium as the objective for robust policy training, ensuring a lower bound on expected returns in two-player zero-sum games.
2.	It introduces acceleration techniques built on previous work, further enhancing performance.
3.	Extensive experiments validate the effectiveness of the proposed algorithm.

**Weaknesses:**

1.	The paper claims that the algorithm converges to Nash equilibrium but provides no theoretical proof, which is unfortunately incorrect. For instance, in a rock-paper-scissors game, each player’s strategy pool includes rock, paper, and scissors. After each update, a player might shift the strategy cyclically (rock to paper, paper to scissors, etc.). Thus, since the strategy pool only retains these elements, the final output is one of the three actions, not the Nash equilibrium (i.e., an uniform distribution over all three choices). The authors also refers to [1] for convergence claims; however, after reviewing [1], it only shows in the proof of their Theorem 2 that the distance to the Nash equilibrium does not increase, —there’s no guarantee of strict reduction, hence no convergence. For example in cases like rock-paper-scissors, where strategies might cycle. This recurrence is commonly seen in Nash equilibrium finding[2].
2.	Numerous established approaches exist in the field of Nash equilibrium finding, including fictitious play, PSRO, no-regret learning, and last-iterate convergence algorithms. However, the paper lacks a literature review of these methods, and the proposed MM-FATROL algorithm does not show clear advantages over them.

[1] Guo, Wenbo, et al. "{PATROL}: Provable Defense against Adversarial Policy in Two-player Games." 32nd USENIX Security Symposium (USENIX Security 23). 2023.
[2] Mertikopoulos, Panayotis, Christos Papadimitriou, and Georgios Piliouras. "Cycles in adversarial regularized learning." Proceedings of the twenty-ninth annual ACM-SIAM symposium on discrete algorithms. Society for Industrial and Applied Mathematics, 2018.

**Questions:**

1.	The proposed method resembles a variant of PSRO; have the authors compared it against this class of methods?
2.	The experimental evaluation across different environments is commendable, but it lacks comparisons with other equilibrium-solving methods, such as PSRO and NSFP. Would additional comparisons with these methods help further validate the approach's effectiveness?

---

### Meta-Review · Area_Chair_Jauu · 2024-12-20

**Metareview:**

The paper focuses on the challenge of training robust policies in two-player adversarial games. The authors propose MM-FATROL, a robust policy training method grounded in the Minimax Theorem, which reduces computational overhead by efficiently identifying promising policy updates. None of the reviewers was enthusiastic about the paper, and there are a few comments about non-rigorous claims and unjustified reasons the proposed method is used over other methods. Moreover the authors did not attempt to address the reviewers' comments on the rebuttal. We recommend rejection

**Additional Comments On Reviewer Discussion:**

The authors did not participate in the rebuttal.

---

### Decision · Program_Chairs · 2025-01-22

Reject